# Sustainable Polypropylene-Based Composites with Agro-Waste Fillers: Thermal, Morphological, Mechanical Properties and Dimensional Stability

**DOI:** 10.3390/ma17030696

**Published:** 2024-02-01

**Authors:** Tatiana Zhiltsova, Jéssica Campos, Andreia Costa, Mónica S. A. Oliveira

**Affiliations:** 1Mechanical Engineering Department, University of Aveiro, 3810-193 Aveiro, Portugal; jessica.e.campos@ua.pt (J.C.); monica.oliveira@ua.pt (M.S.A.O.); 2TEMA—Centre for Mechanical Engineering and Automation, University of Aveiro, 3810-193 Aveiro, Portugal; 3LASI—Intelligent Systems Associate Laboratory, 4800-058 Guimarães, Portugal; 4OLI-Sistemas Sanitários, S.A. Travessa de Milão, Esgueira, 3800-314 Aveiro, Portugal; andreiac@oli-world.com

**Keywords:** natural fiber-reinforced composites, olive pits, rice husk, non-destructive morphology assessment, polypropylene, short-term water absorption

## Abstract

Natural fiber composites (NFC) are eco-friendly alternatives to synthetic polymers. However, some intrinsic natural fillers’ properties hinder their widespread implementation as reinforcement in polymeric matrices and require further investigation. In the scope of this study, the thermal, rheologic, mechanical (tension and flexural modes), and morphological properties, as well as the water absorption and dimensional stability of the NF polypropylene (PP)-based injection molded composites reinforced with rice husk (rh) and olive pits (op) of 20 wt.% and 30% wt.%, respectively, were investigated. The results suggest that the higher content of the rice husk and olive pits led to a similar reduction in the melt flow index (MFI), independent of the additive type compared to virgin polypropylene (PPv). The melting and crystallization temperatures of the PPrh and PPop composites did not change with statistical significance. The composites are stiffer than the PP matrix by up to 49% and possess higher mechanical strength in the tension mode at the expense of decreased ductility. PPrh and PPop have a superior flexural modulus in the bending mode, while the flexural strength improvement was accomplished for the PP30%rh. The influence of the fibers’ distribution in the bulk of the parts on their mechanical performance was confirmed based on a non-localized morphology evaluation, which constitutes a novelty of the presented research. The dimensional stability of the composites was improved as the linear shrinkage in the flow direction was decreased by 49% for PPrh and 30% for PPop, positively correlating with an increase in the filler content and stiffness. PPop was less susceptible to water sorption than PPrh due to fibers’ composition and larger surface-to-area volume ratios.

## 1. Introduction 

The exponential growth of synthetic polymer production and consumption has resulted in severe problems related to product disposal and depletion of petrochemical feedstocks. Eco-friendly alternative polymer materials are being developed to address environmental and societal concerns. Natural fiber composites (NFC) have garnered substantial interest and extensive research. Such materials have improved environmental behavior and an optimized load-bearing capacity, meeting the quality requirements of polymer components while reducing the reliance on synthetic polymers [1,2,3,4]. 

Agriculture is a significant economic activity in Portugal. However, it generates a large amount of waste that needs to be disposed of properly. Every year, Portugal produces approximately 160,000 tons of paddy rice, generating approximately 20% of this amount as a by-product in the form of rice husk [5]. Another important Portuguese agro-industry is olive production. In 2022, olive production in mainland Portugal amounted to 774,743 tons, which generated significant olive pit residue after oil extraction [6]. The amount of these two industries’ production makes rice husks and olive pits the most abundant agro-industrial waste products in Portugal. Regarding the disposal of these agro-residues, until recently, the primary disposal route was energy generation through incineration, collaterally aggravating environmental pollution [7]. Compared to other agro-waste sources, these fibers are challenging to reutilize, making them unsuitable for animal feeding due to their lack of nutritional value. However, using them as a renewable natural fiber source for the reinforcement of polymer-based composites is a more economical and environmentally friendly alternative [8,9,10,11]. Compared to synthetic analogues, the lower density of natural fibers and their higher toughness than polymer-based matrixes lead to some favorable NFC properties such as a low density and a high mechanical strength, stiffness, and effective thermal and acoustic insulation. Additionally, NFCs have a reduced environmental impact and require fewer resources during processing compared to synthetic fibers [12]. In recent years, a vast body of literature [13] has been dedicated to polyolefin-based NF composites reinforced with rice husk and olive pit fibers. The focus on polyolefins as a choice for the polymer matrix is easy to understand due to their excellent chemical and moisture resistance, low density, good mechanical properties, low cost, and capability to be industrially processed through different methods such as injection molding and extrusion. 

A few challenges are hindering the broad implementation of NFCs. One of the problems arises from the chemical structure of lignocellulosic fibers, having several hydroxyl functional groups at the surface which create a weak interfacial interaction with the functional group in the polymer matrix and promote moisture absorption [14]. The typical way to improve the natural filler/matrix interactions is the addition of coupling agents. Maleic anhydride-grafted polyolefins are the most frequently applied coupling agents and have been investigated in many studies for polyolefin-based composites. Using coupling agents improves the adhesion quality between polymer and filler by reducing the gaps in the interfacial region, simultaneously blocking the hydrophilic groups. The impact of maleic anhydride-grafted polyolefins on NFC quality is multifold, including improvements in dimensional stability [15], decreased water absorption [16,17], and improvements in mechanical properties [3,15,17,18]. In addition, significant research has been dedicated to precisely defining the natural filler content for maximum improvements in NFC mechanical properties. The common ground is an improvement in the elastic mechanical properties, i.e., a higher stiffness at the expense of significant ductility loss due to the insufficient wetting of the fibers by the polymer matrix [2,3,14,19,20,21,22,23]. 

However, where very heavy natural filler loading is concerned, a higher loading may not correlate with a better mechanical performance [1,24].

A few studies address the dimensional stability of NFC [15,25], but none investigate the shrinkage properties of rice husk and olive pit injection molded composites, requiring further investigation on this topic. Moreover, to the authors’ knowledge, no research has been reported on the non-destructive morphology assessment of injection molded NFC, which may be essential for establishing the structure–property relationship. X-ray computed tomography (CT) analyses have been used so far for synthetic composite materials, focusing primarily on assessing the porosity and morphology of fibers [26]. To advance larger-scale industrial implementation of natural fiber composites, both dimensional stability and non-localized morphology evaluations should be addressed.

In light of the above, this study aims to establish the suitability of rice husk and olive pit composites for applications in functional products such as sanitary components, aiming for synthetic polymer replacement and complying with the objectives of the project “OLIpush—Redesign for greater circularity and a smaller environmental foot-print”. For this purpose, the water absorption and the thermal, mechanical, physical, and morphological properties of rice husk and olive pit polypropylene composites with different combinations of fiber loading were investigated. 

## 2. Materials and Methods

### 2.1. Materials

PP-based NFCs with 20 and 30% wt. of rice husk and olive pits, respectively, were prepared on demand by Bio4plas—Biopolímeros, Lda (Cantanhede, Portugal). A total of 1 wt.% of ExxonMobil (Houston, TX, USA) EXXELOR™ PO 1020 Maleic anhydride functionalized polypropylene (PPMA) was added to the compound to improve the matrix/fiber compatibility. The virgin 205CA-40—Polypropylene Random Copolymer by INEOS Olefins & Polymers Europe (PPv) (London, UK) was used as a polymer matrix for the preparation of the composites and for a comparison of their properties. The filler content was chosen to balance the composite’s mechanical performance, processability, and water absorption properties. Higher concentrations of natural fibers might lead, as reported in the literature [1,24], to poor fiber dispersion, reduced compatibility with the polymer matrix, and decreased mechanical properties. 

Before compounding, the rice husk and olive pits were oven-dried for 4 h at 100 °C until they reached approximately 0.7% humidity. The humidity level of the fibers was assessed based on a gravimetric moisture measurement using a moisture balance DAB 100-3 (Kern & Sohn GmbH, Balingen, Germany). The rice husks and olive pits were shredded with an SM 100 Cutting Mill (RETSCH GmbH, Haan, Germany) and sieved through a mesh strainer (mesh size 0.5 mm).

The matrix polymer and additives were premixed, and then the premix was fed at a constant feeding screw velocity of 12 rpm through the 11 mm single feeder (Thermo Electron GmbH, Karlsruhe, Germany) connected to a Process 11—Parallel Twin-Screw Extruder with an 11 mm diameter (Thermo Electron GmbH, Karlsruhe, Germany). The extrusion velocity of the twin co-rotating screw was 70 rpm. The temperature profile of the extruder barrel’s seven zones was set from hopper to die at 165 °C, 170 °C, 175 °C, 180 °C, 185 °C, 190 °C, and 190 °C. The composites, extruded through the 2 mm die, were cooled in a water bath and then pelletized into granules of 2 mm in length.

The composition of the composites and their designations, referred to through the text hereafter, are listed in Table 1. 

### 2.2. Methods

#### 2.2.1. Melt Flow Index

The measurements of the melt flow index were performed using a Göttfert MI-3 machine (GÖTTFERT Werkstoff-Prüfmaschinen GmbH, Buchen, Germany) using the standard ISO 1133-1997 (2.16 kg, 190 °C) [27]. Each NFC lot and virgin PP was analyzed based on five samples to ensure accuracy and reliability. The melt density (p_m_) was calculated as the relationship between the *MFI* (g/10 min) and the *MVR* (melt volume rate, cm^3^/10 min), as shown in Equation (1).
(1)$$\rho_{m}=\frac{MFI}{MVR}$$

#### 2.2.2. Differential Scanning Calorimetry

For the determination of the melting (T_m_) and crystallization (T_c_) temperatures as well as the melting (*H_m_*) and cold crystallization (*H_c_*) enthalpies, differential scanning calorimetry (DSC) was performed on samples weighing approximately 5 to 10 mg. The DSC tests were carried out using the DSC Discovery 250 instrument (TA Instruments, New Castle, DE, USA) following the standard ISO 11357-3 [28] guidelines. TRIOS software (V5.7), the proprietary software of TA Instruments (New Castle, DE, USA) was used for data analysis. To eliminate any thermal history of the material, each material lot underwent two heating and cooling cycles. The data from the second cycle were collected and analyzed. Three samples were analyzed for each material lot. The DSC analysis involved stabilizing each sample at 20 °C, followed by heating it up to 190 °C and then cooling it back to 20 °C at a rate of 10 °C/min. The degree of crystallinity (χ) for each sample was calculated using Equation (2) [23]:(2)$$\chi \left(\%\right)=\frac{H_{m}}{H_{m}^{0}\times f_{pp}}\times 100$$
where *H_m_* (J/g) represents the melting enthalpy of the polymer under analysis. The melting enthalpy of 100% crystalline PP ($H_{m}^{0}$) is known to be 207 J/g [29]. The variable *f_pp_* denotes the weight fraction(wt.%). of PP in the composite.

#### 2.2.3. Processing

Specimens for tensile (ISO 527-1) [30] and flexural testing (ISO 178) [31] were obtained simultaneously through injection molding in the family mold (injection molding machine Euroinj D—065, Lien Yu Machinery Co., Ltd. (Tainan City, Taiwan). Before molding, the composites were dried for 2 h at 85 °C in a hopper dryer connected to the injection molding machine. Between the change in NFC lots, the injection barrel was purged with neat PP. To ensure the material’s purity between the shifting of lots, the first 15 parts were discarded before collecting. The barrel temperature profiles for PPv and the composites are shown in Table 2. The other pertinent injection molding processing conditions are detailed in Table 3.

#### 2.2.4. Density

A density assessment of the produced composites and neat PP was carried out using a hydrostatic method at room temperature on an A&D GH-252 scale (A&D Company, Limited, Tokyo, Japan), according to ASTM D792-13 [32]. Due to the hygroscopic nature of the NFC specimens, they were dried in the oven with forced ventilation at 50 °C for 48 h. Following removal of the specimens from the oven, they were put in a desiccator (HC 200 Humidity Control Cabinet by Guangdong SIRUI Optical Co., Ltd., Guangdong, China) to be brought to laboratory temperature (23 °C). After the specimens were kept in vacuum-sealed plastic bags before the density measurements. Propanol (2-propanol of 99.5% purity by Sig-ma Aldridge (St. Louis, MO, USA) was used as an immersion medium to prevent the samples from floating. The density of propanol at 25 °C is 0.785 g/cm^3^ [33]. The sample size was approximately 28 × 12.7 × 3.2 mm, with a mass of approximately 1.1 g. The density was assessed based on a fivefold replicate for each material lot.

#### 2.2.5. Shrinkage

The composites’ dimensional stability was assessed by evaluating the shrinkage parallel to flow according to ASTM D955 [34] with a bar specimen of 12.7 by 127 by 3.2 mm (Type A). The length of the mold cavity and the parts were measured using digital calipers from Mitutoyo (precision ± 0.01 mm). The average dimensions of five specimens were considered for the shrinkage calculation using Equation (3):(3)$$S_{l}=(L_{m}-L_{s})\times \frac{100}{L_{m}}$$
where S*_l_* is the shrinkage parallel to the flow, %; *L_m_* is the mold dimension parallel to the flow; and *L_s_* is the specimen dimension parallel to the flow.

#### 2.2.6. Mechanical Properties

The Young’s modulus and ultimate tensile strength of the samples were assessed using an X-10kN machine (Shimadzu Scientific Instruments, Columbia, MD, USA) following the ISO 527-1 [30] standard. Ten Type I specimens were subjected to the tensile tests. The experiments were carried out at an ambient temperature in two steps. Initially, the specimens were pulled at a rate of 1 mm/min to calculate Young’s modulus. In the second stage, a 50 mm/min tensile rate was applied and maintained until the specimens ruptured. The data from this second test were used to determine the ultimate tensile strength (σ_u_) and tensile strain at break (ε_b_). 

Flexural testing (3-point bending) was performed at a 5 mm/min speed. The beam-type test specimen with dimensions of 127 × 12.7 × 6.35 mm follows the ISO 178 [31] requirement of 20 ± 1 of the length to the thickness (*h*) ratio. The length span between the supports complies with Equation (4).
(4)$$L=\left(16\pm 1\right)h$$

#### 2.2.7. Morphology Assessment

Morphological analysis of the fractured surface of the tensile specimens was performed using a tabletop scanning electron microscope (SEM) HITACHI TM4000Plus (Hitachi High-Tech Corporation, Tokyo, Japan). The SEM images were acquired at an accelerating voltage of 15 kV. Before imaging, the fractured surfaces of the specimens were carbon-coated under a vacuum for 60 s using an Emitech K950X Carbon Evaporator Sputter (Emitech Group, Montigny-le-Bretonneux, France) to improve the resolution of the micrographs. 

The SEM micrographs provided a morphology assessment of high accuracy; nevertheless, the area under analysis was limited. To identify the internal microstructure of the fiber-reinforced composites and uniformity of the fillers’ distribution along the length and throughout the thickness of this part, micro CT X-ray inspection was used for images acquired with 14 μm resolution. When exposed to X-ray radiation, the internal structure of a part absorbs it at different percentages, which is used to identify the fillers. This was accomplished by analyzing the fragments of the narrow section (13 × 3.2 × 15 mm) of the tensile specimens using a Bruker MicroCt Skyscan 1275 microscope (Bruker Corporation, Billerica, MA, USA). A sequence of projections was obtained and reconstructed using NRECON software v. 2.0 (Micro Photonics Inc., Allentown, PA, USA). CTAN v.1.18 software was used for the initial 2D/3D image analysis and processing (Bruker Corporation, Billerica, MA, USA). The quantitative data were treated with the open-source public domain software ImageJ 1.54h5 [35].

#### 2.2.8. Chemical Composition Analysis

A chemical composition analysis was performed using the Energy Dispersive X-ray Spectroscopy (EDS) module of SEM HITACHI’s TM4000Plus (Hitachi High-Tech Corporation, Tokyo, Japan). EDS was used to determine which chemical elements were present in a sample and estimate their relative abundance. The fractured surface of the tensile specimens was analyzed without any additional coating to avoid interference with the element’s detection.

#### 2.2.9. Water Absorption

A twenty-four-hour immersion test was carried out according to ASTM D570 [36]. The water absorption amount (*WA*) was evaluated by calculating the change in the sample mass to its original mass according to the following Equation (5):(5)$$WA=\frac{\left(W_{t}-W_{0}\right)}{W_{0}}100\%$$
where *W_t_* is the specimen mass at time *t* and *W_o_* is the initial dry mass of the specimen before it is immersed in water. The weighing was carried out within 30 s to avoid errors due to desorption. To ensure statistical significance, eight tensile and flexural specimens of each material lot were tested and designated as TS and FS. The initial dry mass was recorded as follows: before immersion, the specimens were dried in an oven with ventilation for 24 h at 50 °C, cooled in a desiccator, and weighed on an A&D GH-252 scale (A&D Company, Limited, Tokyo, Japan) to the nearest 0.001 g.

## 3. Results and Discussion

### 3.1. MFI and Physical Properties

Polypropylene is one of the lightest polymers, with a density below the density of water. Adding the rice husk and olive pit fibers into the PP matrix resulted in a slight increase in its melt and solid densities, p_m_ and p_s_, respectively, as shown in Table 4. The density increase is to be expected due to the partial collapse of the cellulose cells and the lumen of the fibers under the high pressure of the injection molding [14]. A notable reduction in MFI was observed with the addition of rice husk and olive pits. Compared to PPv (Table 4), the melt flow index of PPrh decreased by approximately 18% and 38% with an increase in the rice husk content. A similar trend was verified for PPop, where the reduction in MFI was approximately 21% and 36%. These findings follow previous research which attributes this phenomenon to the restriction of the polymer matrix flow due to fiber loading [19]. The fibers effectively act as a thickening agent, reducing the fluidity of the PP polymer matrix in the vicinity of the fillers. As can be seen from Table 4, the linear shrinkage of all the PPrh and PPop composites in the flow direction was lower than that of neat PP. The lowest shrinkage was observed for PPrh, constituting a decrease of 49% and 35%, respectively, for 30% and 20% rice husk content, as compared to PPv. Meanwhile, PPop also showed a significant but more moderate reduction in linear shrinkage, down to 23% and 30%, respectively, for 20% and 30% olive pit contents.

### 3.2. Thermal Properties 

Compared to the virgin PP, the inclusion of the cellulosic fillers into the polymer matrix does not alter the melting (T_m_) and crystallization (T_c_) temperatures of the composites in a statistically significant way, independently of the type and content of the fillers, as shown in Table 5. The enthalpy of fusion (H_m_) was slightly lower for the composites compared to the PPv, which is expected, as the amount of PP matrix is lower. However, their crystalline content increased and was directly correlated with the amount of filler, regardless of its type, rising by 23% and 41% for PPrh and 23% and 40% for PPop. These findings corroborate the data reported in the literature [4,23], attributing the rise in crystal content to the interaction between the filler and PP matrix, promoting the nucleation of PP crystals around the rice husk and olive pit particles. A higher crystallinity indicates material stiffness [37], suggesting that all the tested composites possessed a higher elastic modulus and decreased ductility.

### 3.3. Morphology Assessment

Based on the SEM micrographs (Figure 1) of the fractured cross-sections of the tensile specimens, it becomes evident that the distribution of the fibers is not uniform. Additionally, there are indications of clamping, suggesting poor reinforcement/matrix adhesion and, consequently, lower performance of these composites during tensile tests, as seen in Table 6, with lower values for the strain at break compared to the virgin PP. The non-uniform distribution and clamping were more pronounced in the case of olive pit NFCs (Figure 1c,d), where dark areas (voids) resulted from the pulling of the fibers from the PP matrix. In contrast, in the rice husk specimens, it is evident that the NFCs showed a relatively more uniform distribution, particularly in PP30%rh (Figure 1b).

Although the SEM micrographs provide a morphology assessment of high accuracy, they required destructive preparation of the samples, and the area under analysis was restricted. Micro CT X-ray inspection is a valuable, non-destructive technique to obtain insight into the anisotropy of the fillers’ distribution and the granulometry of the incorporated fibers. The fillers are identifiable at the cross sections of the 3D reconstruction of the micro-CT scans (Figure 2). The non-uniformity of the olive pit particles’ distribution throughout the part thickness is evident in Figure 2c,d while the distribution of the rice husk fibers (Figure 2a,b) is more uniform. To quantitatively access the granulometry and fillers’ distribution through the thickness, three cross-section slices of the area, highlighted in the white box, in Figure 2a, were analyzed using the particle analysis module of the ImageJ software. The analyzed scans were designated further in the text as the top, bottom, and middle, where the top and bottom were at 0.2 mm from the part’s surface and the middle indicated its center (Figure 3). 

The parts under analysis were obtained through injection molding, as described previously. As can be depicted from Figure 4a,c,d,f, the clustering of the larger olive pit particles, independent of the fillers’ load, is consistent with the higher viscosity of the polymer melt, leading to larger particle retention in the vicinity of the mold wall. The declared maximum granulometry of the olive pit particles before compounding was 0.5 mm. However, the histogram of the olive pit particles’ size (Figure 5) and X-ray diffraction micrographs (Figure 1) show a significant variation in the particles’ dimensions, with many smaller particles. The latter is to be expected. First, it happens during grinding, then due to a higher mixing intensity during extrusion and injection molding and subsequent friction within the particles, leading to further size reductions and quantity increases, especially evident for granulometry between 15 and 215 µm equivalent diameter (De) (Figure 5). There are more small-size particles in the middle of the part (Figure 5), as well as lesser particle clustering (Figure 4b,e). Due to lower polymer melt viscosity farther from the mold walls, smaller olive pit particles represent lesser resistance to the polymer flow, gravitating to the hotter middle of the part. The latter was more pronounced for olive pit contents higher than 30%. Nevertheless, there were few particles with granulometries ranging from 500 µm to 915 µm, possibly due to some olive pit particle grinding and sieving inefficiency. The particles’ aggregation and a large amount of the dispersed small-size particles leading to a larger interface area may explain the inferior mechanical performance of the PPop compared with PPrh (Table 6 and Table 7) due to poor bonding at the filler–matrix interface region and a decrease in the stress transfer from the matrix to the filler [12].

The rice husk fibers had higher surface-to-area volume ratios and a more irregular shape than the olive pit particles, as shown in Figure 6**.** Thus, their quantitative evaluation was limited to their relative orientation and assessment of the projected area in the different planes throughout the thickness (Figure 7). The rice husk fibers were aligned along the part wall, as shown in Figure 6a,c,d,f. The through-thickness projections of the rice husk fibers at the part surface were more extensive than in the middle (Figure 6 and Figure 7), where the fibers became more randomly oriented (Figure 6b,e) for both PP20%rh and PP30%rh. The alignment of the rice husk fibers along the part wall increased the tensile and flexural moduli (Table 6 and Table 7) and decreased the shrinkage of these composites (Table 4). 

In addition, for PP30%rh, the rice husk fibers’ alignment along the mold wall was conducive to a higher flexural strength (31.44 MPa) compared to PPv (30.94 MPa), corroborating the conclusions of this material’s mechanical performance in the bending mode. The fibers’ alignment near the wall provided better fiber/matrix interlocking under compressive stresses. In addition, the mixing during extrusion and injection molding did not cause a significant reduction in the rice husk fibers’ size or the dispersion of small-size fibers in the PP matrix (Figure 1a,b), contrary to that observed for olive pit particles (Figure 1c,d). It should be noted that despite the declared maximum particle granulometry of 0.5 mm, many rice husk fibers with much larger dimensions were especially evident at the surface layers, sometimes reaching close to 1 mm in length (Figure 6). In addition to some inefficiency in the sieving process, the high aspect ratio and irregular shape of the rice husk fibers may have contributed to this more significantly than specified granulometry variation, as the fibers may have passed through the sieve oriented perpendicularly to their smaller dimension.

### 3.4. Mechanical Properties

As shown in Table 6, the elastic modulus (E) increased in all the composites when compared to PPv, with PP20%rh showing the highest increase (35%) and PP30%op exhibiting a lesser improvement (22%). The lower performance of PPop may be attributed to the olive pit particles’ agglomeration and a large amount of the dispersed small-size particles (Figure 4), leading to a larger interface area of weak PP/filler interaction. Meanwhile, the rice husk fibers had higher surface-to-area volume ratios, which favored interlocking with the PP matrix.

The slight decrease in elastic moduli with an increase in filler content may be explained by an increase in the polymer matrix/filler interface area, as illustrated in Figure 4 and Figure 6, resulting in inadequate fiber-matrix adhesion and non-uniform stress transmission. The overall elastic modulus improvement indicates an increase in the composites’ rigidity, and the difference between the rice husk and olive pit composites may be due to the availability of more area for the rice husk particles to interact with the PP matrix [23,38]. The ultimate tensile strength (σ_u_) also increased, showing a 21% improvement for PPrh independently of the fiber content. The ultimate tensile strength of PPop underwent a more moderate increase for PP20%op, while for the higher filler content (30%), there was almost no difference (1%) when compared to neat PP. Finally, the stiffness of the composites was reflected in a significant decrease in the tensile strain at break (ε_b_), with the specimens rupturing at minimal elongation almost immediately after yielding. The deterioration of these properties can be directly linked to the higher content of the lignocellulosic fibers, their non-uniform distribution, and insufficient fiber wetting by the matrix [7]. The reason for the latter may be associated with several functional groups, mainly the hydroxyl present at the fiber surface, which, in combination with the hydrophobic polymer matrices, results in a weak interfacial interaction [39]. Such a phenomenon, in turn, leads to failure under load. Compared to PPv, the ductility losses were 96% for the NFCs with 20% filler content and 97% for the NFCs with 30% filler content, independently of the filler type (Table 6). The discussed results corroborate the data shared by other researchers [1,4,14,22].

**Table 6 materials-17-00696-t006:** Tensile properties of the materials.

Material	E (MPa)	E↑ * (%)	σ_u_ (MPa)	σ_u_ ↑ * (%)	ε_b_ (%)	ε_b_ ↓ ** (%)
PPv	1020.90 ± 67.00	-	16.28 ±0.74	-	238.2 ±136.3	-
PP20%rh	1377.02 ± 198.2	35	19.65 ± 0.44	20.7	8.55 ± 1.19	96
PP30%rh	1322.58 ± 170.63	30	19.64 ± 0.63	20.6	8.33 ± 1.40	97
PP20%op	1305.80 ± 217.75	28	18.60 ± 0.53	14.3	9.93 ± 1.68	96
PP30%op	1249.57 ± 239.78	22	16.59 ± 0.33	1.0	6.80 ± 1.05	97

* ↑—increase; ** ↓—decrease.

The flexural modulus (E_f_) increased for all the composites, as demonstrated in Table 7. A more pronounced improvement was achieved using a higher concentration of olive pits and rice husk, with the latter showing more potential for enhancing the flexural modulus. Compared with virgin PP, E_f_ increased by 21% and 50% for PP20%rh and PP30%rh, respectively. On the other hand, PPop showed more moderate improvement. These results align with the trend reported by other researchers [3,23]. In the case of flexural strength (σ_f_), a slight decrease was observed for most composites (Table 7), particularly notable for PPop. The deterioration of flexural strengths may be attributed to agglomerations of fillers in the matrix, filler moisture retention, and weak interlocking between the matrix and filler [1]. Meanwhile, there was one exception to this trend: PP30%rh displayed a slight improvement of about 1.6% in flexural strength due to better interlocking of the PP matrix and the rice husk fibers and their alignment along the mold wall (Figure 6). 

**Table 7 materials-17-00696-t007:** Flexural properties of the materials.

Material	E_f_ (MPa)	E_f_ ↑ * (%)	σ_f_ (MPa)	σ_f_ Variation (%)
PPv	1045.30 ± 42.84	-	30.94 ± 0.94	-
PP20%rh	1263.42 ± 77.10	21	30.29 ± 0.83	−2.1
PP30%rh	1572.39 ± 66.97	50	31.44 ± 0.88	+1.6
PP20%op	1170.49 ± 47.33	12	28.61 ± 0.84	−7.5
PP30%op	1356.54 ± 95.49	30	28.97 ± 1.44	−6.4

* ↑—increase.

In addition, an increase in stiffness is conducive to lower shrinkage [15]; therefore, the higher content of the rice husk is directly correlated with a higher modulus and inversely correlated with the shrinkage, which noticeably decreased in comparison to PPv, down to 49% for PPrh (shrinkage 0.67%) and 30% for PPop (shrinkage 0.85%), as shown in Figure 8. The same trend was also verified for PPop but was less pronounced.

### 3.5. Chemical Composition

Figure 9 and Figure 10 show the EDS spectra and the respective zones of the samples under analysis. The samples’ chemical compositions are listed in Table 8. It should be noted that the equipment used did not detect elements with an atomic number less than 3. The X–ray diffraction analysis of PPrh (Figure 9a,b) shows the presence of carbon, oxygen, and silica. The former two elements are present in all lignocellulosic fibers, mainly composed of cellulose, hemicellulose, and lignin, and carbon also originates from the PP matrix (Figure 9c). The presence of silica is a contribution from the rice husk, especially abundant at the protuberances on the outer and the inner epidermis adjacent to the rice kernel, as has been reported by other authors [3,40].

Considering that the primary molecular compounds of olive pits are mainly hydrogen, carbon, and oxygen, in the case of PPop (Figure 10), only carbon and oxygen were detected. As shown in Figure 9b and Figure 10b and Table 8, the amount of oxygen rose with an increase in the fiber load, proportionally to a decrease in the amount of carbon. The same observation was valid for silica, which more than doubled with the increased rice husk content, shown in Figure 9b.

### 3.6. Short-Term Water Absorption

The greatest problem when using lignocellulosic filler in composite materials is its strong sensitivity to water, which is detrimental to their mechanical performance and durability [41]. To explore the potential applicability of the rice husk and olive pit PP-based composites for use in humid conditions, a 24 h immersion test was carried out, the results of which are shown in Table 9 and Figure 11. As was expected, the water absorption in PPv was less than 0.02% due to its intrinsic hydrophobicity. The inclusion of rice husk and olive pit in the PP matrix led to a significant increase in the water uptake, 0.27% for PPrh and 0.17% for PPop, the value increasing with the filler load independently of its type. It must be mentioned that the thicker specimens (FS) showed lower water absorption, as depicted in Figure 11, corroborating the data reported in earlier research [42] regarding to the short-term water immersion. For all the composites, higher water absorption occurred at higher filler loading due to the hydrophilic nature of the cellulosic fibers. The presence of hydroxyl (OH) groups between the macromolecules of the cellulose, hemicellulose, and lignin fiber cell walls promotes the interaction with the water molecules through hydrogen bond formation and reduces the interfacial adhesion between the fiber and the matrix [24,42,43,44,45].

For the rice husk composites, the amount of water absorbed in 24 h was several times lower than reported by Erdogan et al. [3] for PP rice husk composites with similar fiber loads obtained through compression molding. The different processing methods may explain this discrepancy, as the injection-molded specimens were more densely packed when solidifying while constrained under a high pressure. This may have made them less susceptible to water absorption. 

The higher water absorption of PPrh compared to PPop may be attributed to various factors such as filler composition, fiber porosity, and the fibers’ orientations in the PP matrix [36]. Considering that the leading causes of water absorption in cellulosic fibers are, by order of importance, hemicellulose, cellulose, and lignin [43], the amount of hemicellulose in the rice husk is significantly higher than that in the olive pit, as shown in Table 10, promoting the higher water sorption. Moreover, the surface-to-area volume ratios of the rice husk fibers (Figure 6) are larger than that of the olive pits (Figure 4). A larger-interface area of the rice husk particles with the PP matrix and its high porosity [46] leads to a higher water uptake than that of PPop.

## 4. Conclusions

This work has investigated natural fiber composites with endogenous rice husk and olive pit fibers. These composites were processed at a lower temperature than neat PP to avoid degradation of the cellulosic fibers. Their viscosity increased with the reinforcement content but remained within the limits of injection molding processability. The composites became more rigid than virgin PP as their elastic moduli in tension and bending and the tensile stress at break were improved. The flexural strength of the olive pit composites was inferior to neat PP. In contrast, the rice husk composites showed similar flexural strength at 20% reinforcement and a slight improvement at 30%. The dimensional stability of the latter composites was also superior, with the lower linear shrinkage inversely correlated with superior elastic properties and higher crystallinity. The outstanding performance of the rice husk composites was due to the intrinsic fiber properties: a high surface-to-area volume ratio; the fibers’ alignment near the part walls, providing fiber/matrix interlocking under compressive and tensile stresses; and more uniform distribution of the particles compared with the olive pit composites. However, the same rice husk fiber properties that granted superior mechanical performance made the respective composites more prone to water absorption. 

In conclusion, both composites have the potential to be a viable eco-friendly alternative for virgin synthetic polymer replacement for use indoors without excessive exposure to humid air. The most promising in terms of dimensional stability and superior mechanical properties are the rice husk composites, while the olive pit composites are more resistant to hydrolytic ageing. However, no definite conclusion about using these composites in humid conditions may be currently drawn after only the short-term water absorption test, requiring the realization of a long-term water absorption test, which is currently in progress.

## Figures and Tables

**Figure 1 materials-17-00696-f001:**
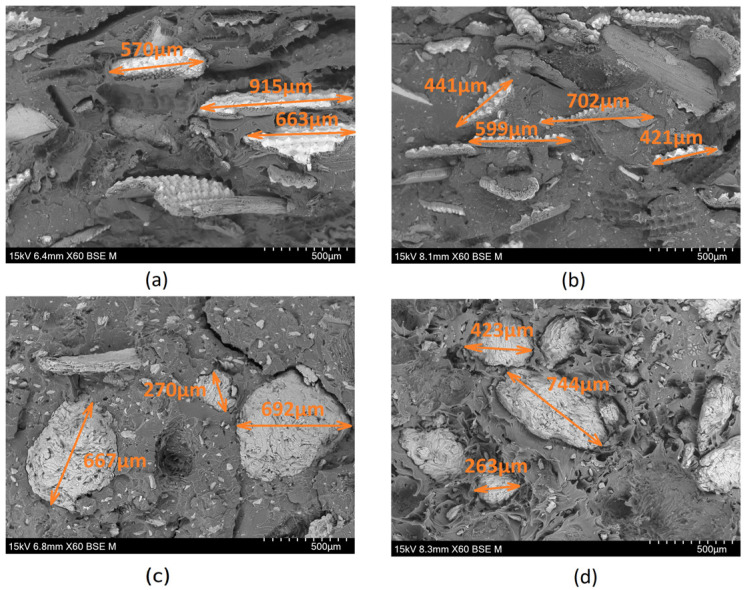
SEM micrographs (60× amplification): (**a**) PP20%hr, (**b**) PP30%hr, (**c**) PP20%op, (**d**) PP30%op.

**Figure 2 materials-17-00696-f002:**
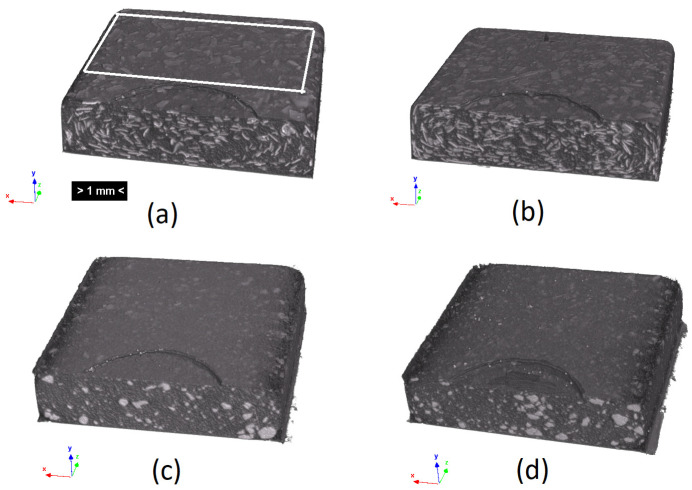
Three-dimensional rendering of the micro CT scans: (**a**) PP20%hr, (**b**) PP30%hr, (**c**) PP20%op, (**d**) PP30%op.

**Figure 3 materials-17-00696-f003:**
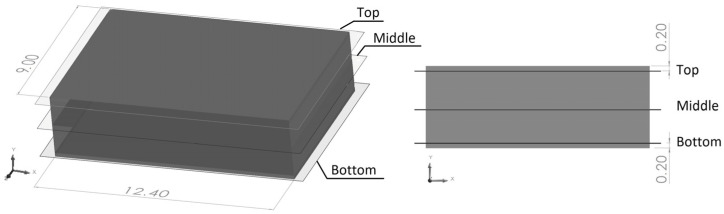
Locations and dimensions (mm) of the area under ImageJ 1.54h5 analysis.

**Figure 4 materials-17-00696-f004:**
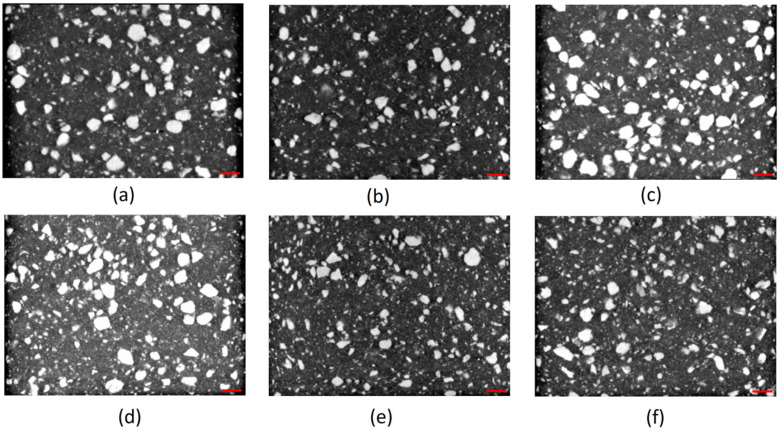
Micro CT scans. PP20%op: (**a**) top, (**b**) middle, (**c**) bottom. PP30%op: (**d**) top, (**e**) middle, (**f**) bottom (red scale bar is equal to 1000 µm).

**Figure 5 materials-17-00696-f005:**
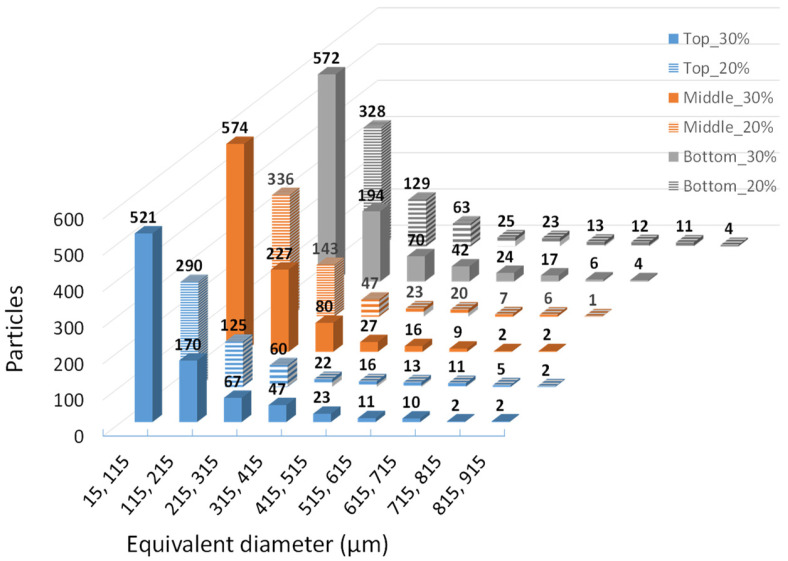
Histogram of the olive pit particle distributions at the top, middle, and bottom scan slices.

**Figure 6 materials-17-00696-f006:**
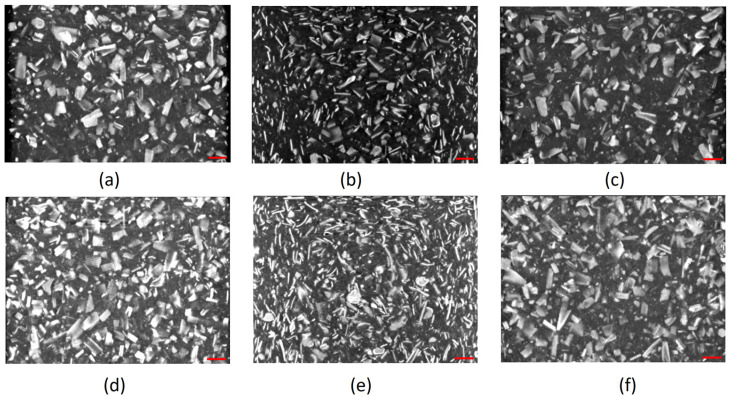
Micro CT scans of PP20%rh: (**a**) top, (**b**) middle, (**c**) bottom PP20%rh; and of PP30%rh: (**d**) top, (**e**) middle, (**f**) bottom, (red scale bar is equal to 1000 µm).

**Figure 7 materials-17-00696-f007:**
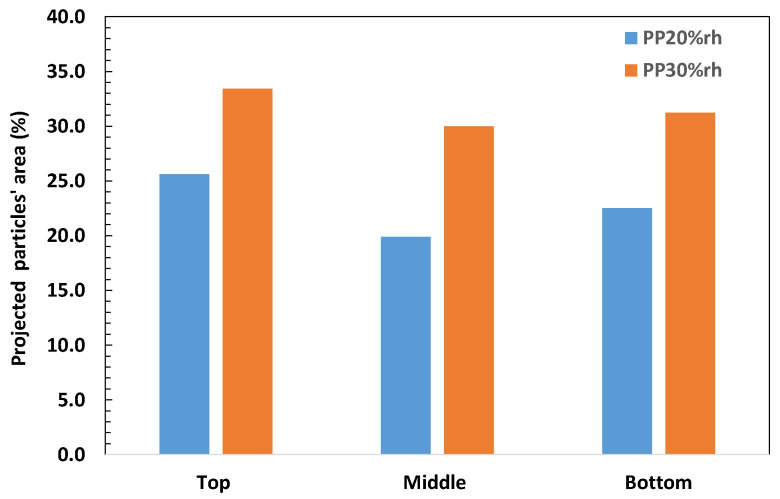
Areas of projected rice husk particles at the different parts’ cross-sections.

**Figure 8 materials-17-00696-f008:**
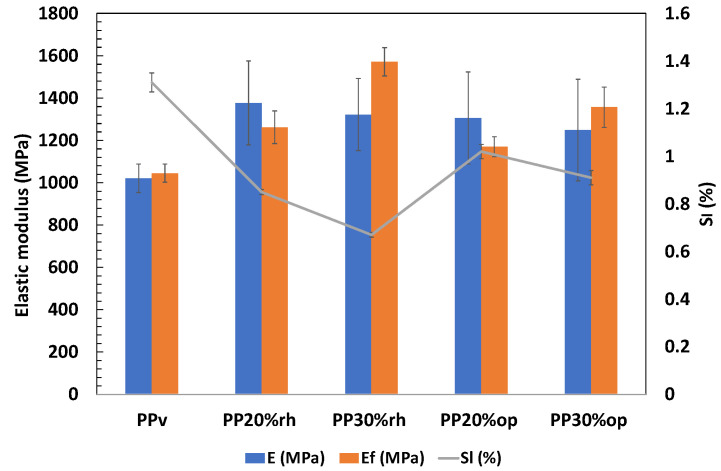
Tensile (E) and flexural (E_f_) elastic moduli vs. linear shrinkage (S_l_) of the composites.

**Figure 9 materials-17-00696-f009:**
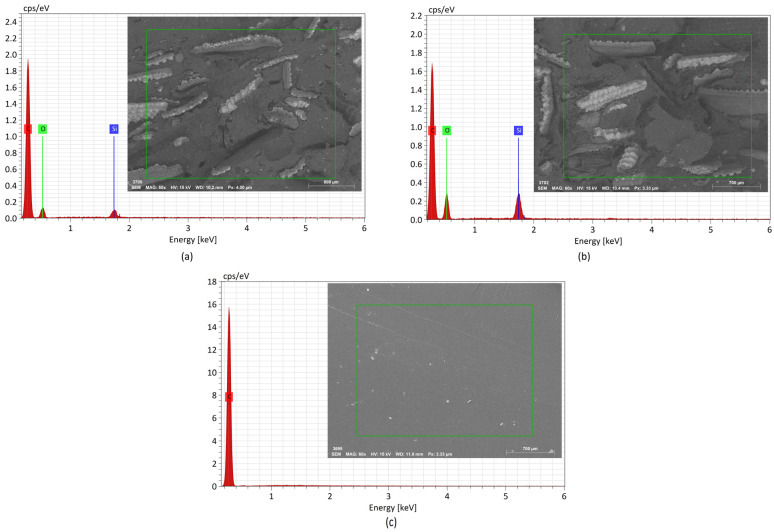
EDS spectra and the respective zones (green contour) of the samples under analysis: (**a**) PP20%rh, (**b**) PP30%rh, (**c**) PPv.

**Figure 10 materials-17-00696-f010:**
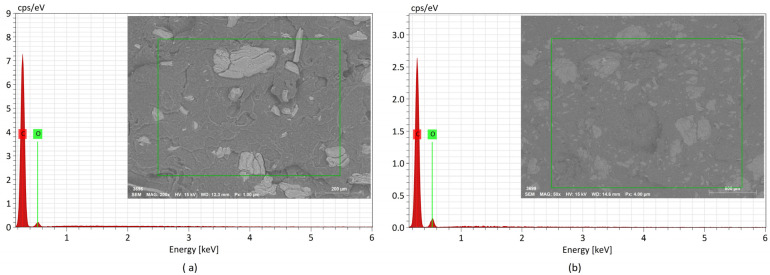
EDS spectra and the respective zones (green contour) of the samples under analysis: (**a**) PP20%op, (**b**) PP30%op.

**Figure 11 materials-17-00696-f011:**
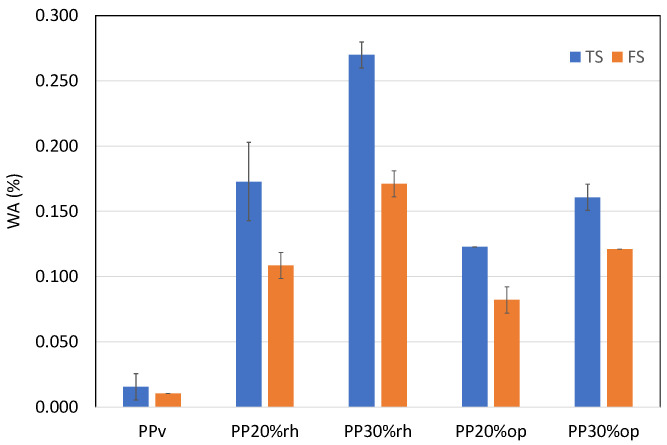
Short-term (24 h) water absorption.

**Table 1 materials-17-00696-t001:** Material compositions.

Designation	Composition (%)			
	PP	Rice Husk	Olive Pits	PPMA
PPv	100	-	-	-
PP20%rh	79	20	-	1
PP30%rh	69	30	-	1
PP20%op	79	-	20	1
PP30%op	69	-	30	1

**Table 2 materials-17-00696-t002:** Injection molding machine barrel temperature profile.

Barrel Zones	Temperature (°C)	
	PPv	NFC
Nozzle	230	190
Zone 1	220	180
Zone 2	215	175
Zone 3	210	170
Zone 4	205	165

**Table 3 materials-17-00696-t003:** Injection molding processing conditions.

Parameter	PPv/NFC
Mold temperature (°C)	40
Injection velocity (%) *	20
Packing time (s)	40
Cooling time (s)	35

* % of the maximum velocity of the injection molding machine.

**Table 4 materials-17-00696-t004:** Processing and physical properties.

Material	Ƿ_s_ (g/cm^3^)	Ƿ_m_ (g/cm^3^)	MFI (g/10 min)	MFI ↓ *(%)	Shrinkage (%)
PPv	0.824 ± 0.0004	0.722 ± 0.06	12.93 ± 1.50	-	1.31 ± 0.04
PP20%rh	0.888 ± 0.0006	0.781 ± 0.06	10.57 ± 0.64	18.30	0.85 ± 0.01
PP30%rh	0.920 ± 0.0009	0.801 ± 0.05	8.00 ± 0.27	38.20	0.67 ± 0.01
PP20%op	0.891 ± 0.0011	0.813 ± 0.02	10.22 ± 0.29	21.00	1.02 ± 0.03
PP30%op	0.921 ± 0.0011	0.848 ± 0.03	8.23 ± 0.52	36.40	0.91 ± 0.03

* ↓—decrease.

**Table 5 materials-17-00696-t005:** Thermal properties of the materials.

Material	T_m_ (°C)	H_m_ (J/g)	T_c_ (°C)	H_c_ (J/g)	Ҳ (%)
PPv	149.28 ± 0.40	82.61 ± 2.46	118.61 ± 1.22	76.40 ± 2.23	39.9
PP20%rh	149.26 ± 0.17	80.96 ± 4.68	118.77 ± 0.21	74.60 ± 3.81	48.9
PP30%rh	149.29 ± 0.16	81.57 ± 2.87	118.25 ± 0.22	66.00 ± 2.34	56.3
PP20%op	149.36 ± 0.12	81.50 ± 0.75	119.45 ± 0.10	75.26 ± 0.41	49.2
PP30%op	149.28 ± 0.40	82.61 ± 2.46	118.61 ± 1.22	76.40 ±2.23	56.0

**Table 8 materials-17-00696-t008:** Chemical composition of the materials.

Material	Element Mass (%)		
	C	O	Si
PPv	100	-	-
PP20%rh	82.41	16.17	1.423
PP30%rh	72.06	24.62	3.32
PP20%op	91.03	8.97	-
PP30%op	84.91	15.09	-

**Table 9 materials-17-00696-t009:** Short-term (24 h) water absorption.

Specimen Type			WA (%)		
	PPv	PP20%rh	PP30%rh	PP20%op	PP30%op
TS	0.015 ± 0.01	0.173 ± 0.03	0.270 ± 0.01	0.12 ± 0.00	0.161 ± 0.01
FS	0.010 ± 0.00	0.109 ± 0.01	0.171 ± 0.01	0.082 ± 0.01	0.121 ± 0.00

**Table 10 materials-17-00696-t010:** Rice husk and olive pit compositions.

Filler	Cellulose (%)	Hemicellulose (%)	Lignin (%)	Ref.
Rice husk	42.80	37.20	22.50	[7]
Olive pit	48.62	16.78	33.67	[47]

## Data Availability

All data generated or analyzed during this study are included in this article.

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
