# Peer review of "Sustainable Polypropylene-Based Composites with Agro-Waste Fillers: Thermal, Morphological, Mechanical Properties and Dimensional Stability"

_materials, 2024, doi:10.3390/ma17030696_

Round 1

Reviewer 1 Report

Comments and Suggestions for Authors

The article explores the morphology and the impact on mechanical properties of rice husk (RH) and olive pits (OP) in a polypropylene (PP) matrix. The structure of the article is comprehensive, and it has some instructive significance for the utilization of agro-waste. The specific suggestions are as follows:

1. **Table 1:** The entry labeled as "3" should be corrected to "30."

 2. **Line 284, Line 287, and Table 6:** The format of Ef needs to be standardized, and it is suggested to use subscripts for all occurrences of f. Also, consider standardizing the format of δf.

 3. **Figure 1:** The green scale in the figure is not clear. It is recommended to change the color, increase its size, and improve its clarity.

 4. **Micro CT scans:** While micro CT scans effectively reflect internal structures, it is suggested to standardize the statistical methods for rice husk and olive pits. It is recommended to use the same statistical method in Figure 7 as in Figure 5 for easier comparison. This will make the particle size of rice husk clearer.

 5. **Figure 5:** The title and scale on the X-axis are unclear. Is the X-axis title ranging from 15 to 915? Why is the title divided into two lines? It is suggested to use "equivalent diameter" as the X-axis title instead of "De."

 6. **Figure 8:** The discussion of Figure 8 is best placed in section 3.3 "Mechanical properties." To clarify the mechanical properties while introducing the morphology, it is suggested to first write section 3.4 "Morphology Assessment" and then proceed with section 3.3 "Mechanical Properties."

Reviewer 2 Report

Comments and Suggestions for Authors

This article seems interesting after a preliminary lecture, but it becomes terrific when a profound analysis is performed. This reviewer has found a myriad of inconsistencies and faults that make it impossible to accept this article in any scientific media. The first concern that impacted this reviewer is that the authors have used commercial composites and compared them to a virgin polymer supplied by another company, without providing any property to justify this election. Well, this is absurd since it is a polymer science viewpoint and denoted that the authors have poor knowledge of polymeric system behavior. In other words, the authors are comparing pears and apples. Consequently, all the discussions emerging from the obtained results are merely spurious and have an absolute lack of scientific sense.

In addition to this, there are plenty of questions related to the preparation of the materials and information about the materials, the characterization section, the absence of degradation studies during processing (the main problem of this type of composite), the inappropriate use of some techniques, and a long list beyond that makes impossible this article to be published anywhere.

Consequently, this article must be REJECTED.

Reviewer 3 Report

Comments and Suggestions for Authors

The manuscript titled "Sustainable polypropylene-based composites with agro-waste fillers: thermal, morphological, mechanical properties, and dimensional stability" investigates the properties of polypropylene composites reinforced with rice husk and olive pits. The study aims to improve the environmental sustainability of polymers by using agricultural waste as fillers. It explores various aspects of these composites, including their thermal behavior, mechanical strength, water absorption, and dimensional stability. The findings suggest that adding agro-waste fillers to polypropylene can enhance certain properties like stiffness and dimensional stability while also presenting challenges like reduced ductility and increased water absorption. The research contributes to the development of eco-friendly materials with potential applications in functional products.

The manuscript is well-structured. The authors did an excellent job expanding the subject in the introduction section with an acceptable amount of recently published works. The experimental section is well-established, and the results layout is good with various techniques. The morphology section, for example, is well-done. I recommend publication after minor revisions:

·       There is a typo in Table 1: The percentage of olive pits should be 30 and not 3.

·       From the beginning I was looking for justification for why the authors chose 20% and 30% of the natural fibers. I would like to think that this was to have a balance between enhancing the composite's properties and maintaining its processability and overall performance. Higher concentrations of natural fibers might lead to issues like poor fiber dispersion, reduced compatibility with the polymer matrix, and difficulties in processing. These percentages are possibly chosen based on preliminary studies or literature that indicate optimal performance in terms of mechanical strength, thermal stability, and other relevant properties of the composite material. Proper explanations need to be included.

·       The values in Table 4 (and other places) should be written in the correct form with the subscription:          Tm à Tm        Hm à ΔHm     Tc à Tc           and so on

Reviewer 4 Report

Comments and Suggestions for Authors

I will mainly comment on the results related with mechanical properties.

The abstract resumes the article's content correctly but must stress its novelty.

The first sentence is not part of the research and without an LCA cannot be affirmed.

I’m not a native English speaker, but I’ve seen more usually agro-residue than agro-residual.

Line 54, Low density is relative. Add information about compared with what the density of natural fibers is low. Similar comment with the strength and stiffness.

Line 57, add references.

Did the authors use an extensometer?

Ensure that all the acronyms are defined.

The density increase is to be 228 expected due to partial collapse of the cellulose cells under high pressure of the injection 229 molding”  The increase  the increase can be also due to the collapse of the lumen of the fibers.

Please define ps and pm in Table 3.

The lack of a lineal behavior on moduli vs %of reinforcement can be indicative of bad dispersion of the fibers inside the polymer.

Table 5. Please add the strain of the matrix at its highest strength. Usually, the composites break at such stress and can be useful for comparison purposes.

Rh shows higher aspect rations than op, and that can cause the differences in strengthening abilities between this reinforcements.

Round 2

Reviewer 2 Report

Comments and Suggestions for Authors

After reading the authors' comments, this reviewer ascertains that they are not very likely to receive criticisms of their poor "technical report." In any case, the response received is far from solving any of the concerns proposed by this reviewer. If any, it is the fault of this reviewer to name commercial to the composites provided by a supplier, but just so. This fact does not invalidate that the authors are just comparing a composite to a virgin material; obviously, all the properties must be different. So what? What are the scientific seeds of this practice?

Unfortunately, this reviewer, with more than 30 years of expertise in vegetal-filled composites, opines that the article is a very partial technical report with some properties, avoiding including characterization related to the degradation of the polymers and the accurate content of the filler. Note that the nominal filler content and the actual are very often very different when the processing method is extrusion, so the authors must check these aspects to have a robust "technical report," which is what the article can aspire to be as the most.

Concerning the second question: "In addition to this, there are plenty of questions related to the preparation of the materials and information about the materials, the characterization section, the absence of degradation studies during processing (the main problem of this type of composite), the inappropriate use of some technique.", I am afraid that the authors have not understood the concerns. The criticism was related to the numbers and type of characterization technique used rather than the description. This reviewer has the impression that the authors have used the ones they have access to but not the most important ones to determine the fundamental behavior of the system. The absence of any degradation study (not even a TGA) is very strange to this reviewer. Here, the authors provide a citation to support that below 190ºC, there is no degradation. This reviewer was entirely surprised and has read this reference carefully. The main difference he has found (apart from having used different fibers well characterized) is that this article has used a lower extrusion profile (140-145-150-155ºC) than the authors (165-170-175-180-185-190-195ºC) and very other injection conditions (reference profile: 140-150-155-160ºC) versus the just 190ºC in dye reported by the authors. Let me say that such differences in temperature, jointly with the shear and elongational forces during processing operations, may cause severe degradation in the fillers and induce degradation in the PO.

Let me recommend being honest when providing a reference to support any concern. The one you have cited is not valid for the authors' purpose, as this reviewer has just demonstrated in the previous paragraph.

So, in light of the responses by the authors, this reviewer strongly recommends rejecting the article.
